# Two-Stage Prototypical Networks Reveal Mosquito Flight Patterns

## Abstract

Understanding behavioral movements in mosquitoes is fundamental for monitoring arbovirus transmission. Most existing Artificial Intelligence (AI) methods recognize tiny insects as background and fail to extract correct features from video frames. To address this issue, we propose a two-stage few-shot classification by Movement Density Map (MDM) prototyping called two-stage prototypical networks. A novel approach that integrates object detection with two-stage prototype training to analyze and identify mosquito behavior from videos. In the first stage, mosquitoes are detected using a fine-tuned YOLO, achieving a maximum mean Average Precision (mAP50) of 97.8%. The detected areas with eliminated backgrounds are then aggregated into MDMs. This mechanism enables encoding hundreds of frames into a single spatiotemporal representation that reveals biologically meaningful flight patterns over time. The MDMs are then mapped into a Vision Transformer (ViT) embedding environment, where class-level prototypes are generated for few-shot classification under 1 and 5 exposures using prototypical networks. Results on datasets of dengue and Zika-carrier mosquitoes, as well as non-carrier ones, collected over 13 days and nights show that our approach significantly extracts more accurate features than a common single-stage prototypical network, leading to an overall performance accuracy of 85.86%. These findings reveal that two-stage prototyping is a reliable and scalable solution for analyzing tiny-object biological videos and holds promise for other spatiotemporal recognition tasks where motion aggregation is critical.

## 1 Introduction

Analyzing complex behavioral patterns in videos has recently become an important research frontier for machine learning based on AI. The challenges of this task become more apparent when the objects are very small, move chaotically, and occupy only a small portion of each frame. Identifying infected mosquitoes is an example of this challenge. Mosquitoes act as an acute function in the transmission of arboviruses, such as dengue and Zika, infecting millions of people worldwide each year (Bhatt et al., 2013; Achee et al., 2015; Kraemer et al., 2015; NODEN et al., 2015). Standard monitoring approaches, including morphological measurements, are unable to accurately capture behavioral changes in mosquitoes that result from viral infections. Biological research proposed that viruses can affect the neural activity and movement patterns of mosquitoes, causing movements in flight dynamics that cannot be observed and measured by standard approaches (Carvalho et al., 2015; Gaburro et al., 2018). This perspective advances the design of computational approaches that can transform raw movement data into meaningful indicators of infection status. Recent advances in computer vision have highlighted the key role of deep learning in automating several entomological tasks (Goodfellow et al., 2016). High-resolution imaging and Convolutional Neural Network (CNN)-based approaches have been applied to egg detection (Javed et al., 2023a), tracking and quantifying mosquito flight directions (Javed et al., 2023b), and species classification in intricate environments. For example, various types of Mask R-CNN have been able to identify objects with very high accuracy even in noisy environments. Also, systems based on convolutional neural networks, such as EggCountAI, have significantly outperformed traditional counting methods achieving high accuracy and scalability. These achievements show that machine vision technology is capable of extracting important and interpretable patterns from complex and noisy data. However, the use of these advances to distinguish between infected and control mosquitoes using behavioral videos remains

largely unknown. From a machine learning perspective, the main obstacle is the scale imbalance and background dominance of the data. Unlike still images, videos have spatiotemporal complexities and require representations that focus on movement patterns rather than reflecting background features. In typical mosquito videos, the insects occupy only a small portion of the image, while fixed elements such as the cage and background lighting conditions make up the majority of the pixels. CNNs typically absorb these unnecessary features, resulting in samples that reflect background variations rather than true, meaningful patterns of biological behavior. As a result, even advanced multi-stage algorithms achieve high accuracy without providing a valid model of mosquito behavior. This bias suggests that novel approaches should directly suppress background features while accurately preserving the spatiotemporal footprint of mosquito movements. Few-shot learning is an efficient and attractive approach when large, labeled datasets are not available. Prototype networks have become a cornerstone in this field by mapping data into a latent space and forming prototypes for each class (Snell et al., 2017). However, the standard architecture of these networks is based on the assumption that distinguishing features can be extracted from single frames or short clips. This assumption is rarely true for small object behavioral data. In fact, diagnostic clues are usually hidden in the long-term aggregation of motion paths across the entire video. This mismatch between the model's induced bias and the actual nature of the task severely limits the applicability of conventional few-shot learning methods to classify mosquito behavior. One promising approach is the use of density maps, a tool that has previously shown its effectiveness in diverse areas such as population counting (Zhang et al., 2016) and video-based density estimation (Hossain et al., 2020). The key idea of this approach is that compressing spatiotemporal activities into a density representation can remove noise while preserving the original motion patterns. Accordingly, we proposed using Movement Density Maps (MDM) as prototypes in the analysis of mosquito behavioral videos. By aggregating detection results over time and transforming them into Gaussian-smoothed heat maps, these maps summarize the overall distribution of insect movement paths and provide a compact yet interpretable descriptor of group-level behavior. Based on this approach, we propose a two-stage prototyping framework for video classification. In the first step, a set of YOLO detectors is used to identify the locations of mosquitoes and remove or mask unnecessary background parts (Diwan et al., 2022; Redmon et al., 2016; Shafiee et al., 2017). Then, the obtained coordinates are converted into density maps and, by averaging over the video, an MDM prototype is created that represents the overall behavioral pattern. In the second step, these prototypes are transferred to the embedding space through an attention-based transformer and then fed into prototyping networks to build prototypes at the class level. This design makes the induced bias of few-shot learning consistent with the biological realities of behavioral changes caused by infection; this means that the main importance lies not in the appearance of a single frame, but in the spatio-temporal pattern of movement as a whole. Few-shot learning is particularly necessary in our setting because labeled mosquito flight videos under controlled infection conditions (Zika, Dengue, Control) are extremely limited. Due to biosafety and the difficulty of collecting high-quality infection-specific recordings of insects, only a few samples per class are available. Therefore, a method capable of generalizing based on very few labeled samples is essential for this case study. Current few-shot methods, such as conventional prototypical networks, lead to wrong embedding features for tiny biological objects. Thus, we need a reliable framework that can detect and classify the correct behavior of tiny objects. In this regard, we introduce two-stage prototypical networks that can distinguish the authentic behavior of tiny objects in videos under few-shot learning conditions reliably. Our proposed framework has three main advantages over conventional approaches. First, by directly removing background segments, one of the main sources of bias in mosquito video analysis is eliminated. Second, by compressing hundreds of frames into a single MDM, temporal redundancy is reduced while meaningful behavioral trajectories are highlighted. Third, generating biologically interpretable prototypes enables multi-step generalization in situations where labeled data are limited. Importantly, our approach bridges two previously separate research areas: first, investigating the neurotropic effects of arboviruses on mosquito behavior (Carvalho et al., 2015; Gaburro et al., 2018) and second, learning density-based representations in machine vision (Hossain et al., 2020). In summary, the main achievements of this research can be summarized as follows:

1. Provide prototypes of MDM as an innovative and efficient representation method for tiny object video classification in biomedical fields.

2. Propose a two-stage sampling framework that integrates YOLO-based detection with density map compression and the use of sampling networks.

3. Improve the accuracy of multi-stage classification between dengue and Zika virus-infected mosquitoes and control (non-infected mosquitoes) and surpass conventional single-stage approaches in terms of efficiency and feature correctness.

4. Demonstrate the generalizability of the introduced framework to other areas of small object video analysis, beyond the scope of entomology.

Relying on the integration of entomological knowledge and new machine learning technologies, these findings not only advance the level of multi-stage video classification but also provide a new perspective on the role of subtle behavioral changes as computational biomarkers.

## 2 RELATED WORKS

### 2.1 FEW-SHOT LEARNING WITH PROTOTYPICAL NETWORKS

Few-shot learning has been widely studied as an effective approach for classification in situations where training data is limited. Prototype networks, first introduced by (Snell et al., 2017), learn a metric space in which each class is represented by the average of its embedded support instances. In this approach, the classification process is performed by measuring the distance of new instances to the prototypes of each class. Their proposed approach achieved advanced results on the Omniglot and miniImageNet datasets, showing that a simple inductive bias can outperform more complex meta-learning models.

Another research, such as Matching Networks (Vinyals et al., 2016), Optimization-Based Meta-model Learning (Ravi & Larochelle, 2017), and Model-Independent Metamodel Learning (MAML) (Finn et al., 2017) have emphasized concepts such as episodic learning, gradient-based adaptation, and metric learning. These approaches are particularly important in the analysis of mosquito behavioral videos, where labeled data is very limited and learning stable representations from a small number of examples plays a key role in improving generalizability.

### 2.2 DENSITY-BASED REPRESENTATIONS IN VIDEO ANALYSIS

Density map-based representations have been successfully applied to tasks such as population counting and crowd density estimation. Early regression approaches attempted to model the overall count using artificial features (Bhatt et al., 2013; Lempitsky & Zisserman, 2010), but these methods were not robust enough in highly congested conditions.

In this regard, (Zhang et al., 2016) provided a multi-column convolutional neural network (MCNN) for population counting. This model generates density maps by combining input fields of different sizes, thus overcoming the scaling problem. They validated the method on the ShanghaiTech dataset (330,000 tagged heads) and showed that the density maps not only estimate the total number of individuals but also preserve their spatial distribution. They also managed perspective distortions better by using adaptive geometric kernels and adapted the MCNN to new domains with limited tagged data through transfer learning strategies. Such insights are also valuable for modeling mosquito flight paths, where motion scaling and overlaps pose similar challenges to human crowd images.

Since the introduction of the Multi-Column Convolutional Neural Network (MCNN), more advanced models have been proposed to improve the quality of density maps. For example, (Sam et al., 2017) introduced the Switch-CNN method, which better handles the scalability problem by dynamically routing and automatically selecting between specialized regressors. Also, (Sindagi & Patel, 2017) developed the Contextual Pyramid CNN, which combines local and global cues to produce more accurate and higher-quality density maps.

In addition, several fundamental approaches have paved the way for further developments. (Idrees et al., 2013) introduced multi-source and multi-scale features for counting in very dense crowds, while (Zhang et al., 2015) investigated the transfer of convolutional networks between different scenes for crowd analysis. Also, (Rodriguez et al., 2011) presented a hybrid model of detection and density estimation, and (Lempitsky & Zisserman, 2010) proposed a density regression approach based on dense SIFT features. These studies paved the way for more advanced models, including the Composition Loss framework presented by (Idrees et al., 2018) that simultaneously optimizes counting, density map estimation, and localization in dense crowds.

In addition, significant progress has been made in the video domain. For example, (Hossain et al., 2020) introduced a multiscale optical flow pyramidal network that achieved improved performance

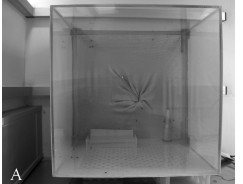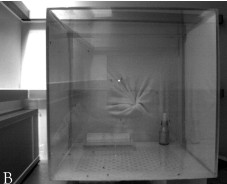

Figure 1: Two sample frames of recorded videos regarding non-infected mosquitoes in A) daytime and B) nighttime.

on video crowd datasets by combining spatiotemporal information with CNN-based density estimation. They showed that compressing movement cues into density maps increases the robustness of the model against obstruction and noise. Such insight is particularly valuable in the analysis of mosquito behavioral videos, where motion data plays a key role in recognizing behavioral patterns. Taken together, these studies demonstrate that density map-based representations are powerful tools for modeling aggregate movement patterns in the presence of obstruction and noise. This capability makes them directly valuable for analyzing the collective movement of mosquitoes and investigating the density of their flight paths over long timescales.

## 2.3 DEEP LEARNING IN ENTOMOLOGICAL VIDEO ANALYSIS

Recently, computer vision has also been applied to mosquito-related challenges. (Javed et al., 2023a) developed EggCountAI, a convolutional neural network-based system for counting Aedes aegypti eggs that achieved accuracy above 98%, even in conditions of overlapping or clustered eggs. This method performed better than traditional tools such as ICount (Gaburro et al., 2018) and MECVision (Kittichai et al., 2024), as well as older approaches based on image separation or wavelet transform (Wan Yussof et al., 2018).

In the area of behavioral studies, (Javed et al., 2023b) used convolutional neural networks to track the flight dynamics of mosquitoes and showed that deep models are able to capture subtle changes in spatiotemporal movements. Additionally, the use of extended versions of Mask R-CNN and spline interpolation methods enabled robust and long-term tracking of Aedes aegypti behavior in crowded environments.

Beyond purely image-based approaches, virological studies in entomology have shown that infection can directly affect mosquito movement. For example, (Gaburro et al., 2018) showed that the Zika virus alters neuronal activity in Aedes aegypti, causing hyperexcitability and abnormal locomotor activity. These findings reveal the interplay between pathogen-induced neural regulation and observable locomotor dynamics.

Collectively, these studies chart a path from static, traditional egg counting to dynamic approaches based on density maps and few-shot learning in mosquito monitoring. Using density-based localization frameworks (Zhang et al., 2016; Idrees et al., 2018), the future video analysis systems in entomology will be able to provide a stable, long-term, and infection-sensitive model of mosquito behavior.

## 3 DATASET

Data collection was conducted in a cubic cage containing 15 mosquitoes. A sugar-water source was provided inside the cage as nourishment. Recordings were obtained over a period of 1 to 13 days, encompassing both daytime and nighttime conditions. A camera was positioned in front of the cage to continuously capture the behavioural activities of the mosquitoes during these periods. Experiments were performed for non-vector mosquitoes as well as for dengue and Zika vectors. Accordingly, the resulting dataset is organized into six classes: dengue-infected (day/night), Zika-infected (day/night), and non-infected mosquitoes (day/night). Figure 1-A illustrates representative frames recorded during daytime, while Figure 1-B presents corresponding examples captured at night. As shown in Figure 1, environmental factors, particularly lighting variations between day and night, uneven illumination across the left and right sides of the cage, and the camera's diverse angle, may introduce a wrong embedding space, consequently affecting classification validity.

Figure 2: A)Visualization of extracted features for embedding space. The first row illustrates a raw frame after applying GradCam and the attention map, while the second row indicates applying them after making an MDM. B) The procedure of the two-stage prototypical network.

## 4 METHODOLOGY

The main methodological contribution of this work is the design of a two-stage prototypical network for modeling fine-grained behavioural cues of tiny moving objects in video. While embedding extraction is a standard component of conventional prototypical networks, the construction of Movement Density Maps (MDMs) and the use of prototype-level representations are essential in our setting. Conventional prototypical networks often produce embeddings dominated by background appearance rather than subtle motion cues, leading to unreliable representations for extremely small biological objects. In contrast, MDM prototypes aggregate mosquito movement patterns across entire videos, mitigating background bias and preserving behaviour-related dynamics. Therefore, combining MDM-based prototyping as the first stage and embedding-based prototyping as the second stage provides a robust, generalizable solution for few-shot classification under severe data scarcity.

In conventional few-shot learning, a one-stage prototyping approach has been proposed by Snell et al. (Snell et al., 2017), which is an effective technique for few-shot classification. However, in terms of classifying biomedical videos by the movement of tiny objects, neural networks extract features from video frame backgrounds that can create an incorrect embedding space for few-shot classification, particularly when the movement of tiny objects, such as insects, is the most substantial hallmark for video classification. Under these circumstances, the embedding space contains incorrect features, and few-shot classification is therefore unreliable (see Figure 2-A). To support the claim above, we have created an embedding space using both Vision Transformer and ResNet models. Then, we visualize the extracted features of the embedding space using explainable AI techniques, such as attention maps. To tackle this challenge, we propose a novel few-shot classification pipeline called two-stage prototyping that can reliably extract the correct features from tiny insect videos and can be trained with a limited amount of data. The two-stage prototyping approach consists of two main steps for creating prototypes. In the first stage, insect movements are detected and captured by MDM prototypes. The second step involves producing and classifying prototypes for each class, as previously introduced as prototypical networks with upgraded embedding and loss functions. It should be noted that in this study, prototyping refers to passing tensors through a neural network or mathematical function and calculating their mean to produce the centroid of these tensors. The following is a description of how the two-stage prototyping is developed. Alos, Figure 2-B depicts the whole procedure.

### 4.1 THE FIRST STAGE OF PROTOTYPING (MDM STRUCTURE):

The videos are recorded over several minutes, in which mosquitoes stay mostly invisible to any scale or low contrast-based appearance features. This, therefore, requires representations at the level of behavior rather than a typical feature extraction from pre-trained models. Thus, the model should provide correct representation of the insect movements within videos.

Conventional embedding methods extract a myriad of features from video frame backgrounds while the classification target is recognizing mosquito behavior. Thus, we need to make sure that no features are extracted from the background to feed the embedding space in this step. To capture the motion behavior of mosquitoes throughout the whole video, we use frame-level detections to build

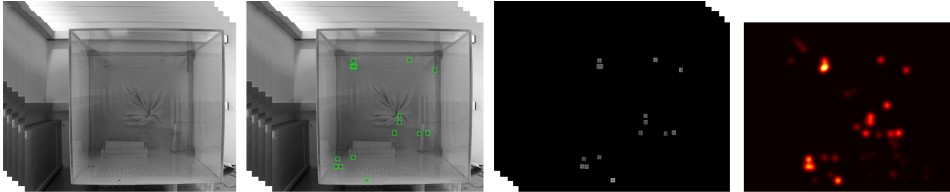

Figure 3: MDM creation procedure. The figure shows raw frames after applying object detection and background elimination, and then creating an MDM.

a single map known as MDM. For each frame t, YOLO returns a bounding box centroid. These coordinates are then projected into a static spatial grid of size H×W. Each point is encoded using a Gaussian kernel to make MDM images.

To begin with, a few initial frames of a video are cut to annotate and fine-tune the YOLO model. Then, YOLO recognizes mosquitoes and specifies regions occupied by them with equation 1 (Redmon & Farhadi, 2018):

$$
\begin{aligned}
b_x &= \sigma(t_x) + c_x \\
b_y &= \sigma(t_y) + c_y \\
b_w &= p_w e^{t_w} \\
b_h &= p_h e^{t_h}
\end{aligned}
\tag{1}
$$

Here, $(b_x, b_y)$ and $(b_w, b_h)$ denote the predicted box center and size. $(t_x, t_y, t_w, t_h)$ are YOLO raw outputs, $\sigma(.)$ is sigmoid, $(c_x, c_y)$ are grid-cell offsets, and $(p_w, p_h)$ are anchor box priors following the standard YOLO parameterization.

Following the detection of tiny objects in the video, all the background is set to zero (equation 2) except for the pixel areas recognized by YOLO. In this way, the model conserves only the locations of mosquitoes in the video frames and eliminates the whole left background. Then, non-zero pixels are converted to adapted Gaussian pixels as calculated by equation 2 (Idrees et al., 2018) since it depicts the presence rate of mosquitoes during the video, which means their movements over time.

$$
D(x, y, f(.)) = \sum_{i=1}^{n} \frac{1}{\sqrt{2\pi f(\sigma_i)}} \exp\left(-\frac{(x - x_i)^2 + (y - y_i)^2}{2f(\sigma_i)^2}\right)
\tag{2}
$$

In equation 2, $(x_i, y_i)$ are mosquito locations detected by YOLO, n is the number of detections, and $f(\sigma_i)$ maps each detection's scale to the Gaussian kernel width. D(x,y) represents the spatial density of mosquito presence across all frames.

In the next stage, the mean tensor of density maps linked to whole frames of a single video is computed and constructed as a prototype, namely an MDM prototype (equation 3).

$$
M_I = \frac{1}{N} \sum_{I=1}^{N} D(x, y, f(.))
\tag{3}
$$

Here, $M_I$ of sample $I$ is the mean of frame-wise density maps for a video, with N frames in total. This prototype summarizes the full flight behavior for downstream prototypical classification.

Finally, the prototype is then turned into a 3-channel image. This shows the whole movement path of each mosquito as one spatial pattern, with less background variation and more infection-specific behavioral signs. It also allows for few-shot classification because an MDM prototype can be calculated over full video behavior rather than needing the whole video itself. This makes the embedding step quicker, since the pre-trained ViT sets up an embedding space just for a single MDM prototype and not all frames in a video.

Zero-value pixels in an MDM prototype indicate that no insect passed through that region of the frames during the video. Intensities close to zero (not exactly zero) indicate that there were slight movements of insects during the video in those specific locations. The higher the brightness of the pixels in the MDM prototype, the more insects were present in that area of the frames during the video (Figure 3).

## 4.2 THE SECOND STAGE OF PROTOTYPING (UPGRADED PROTOTYPICAL NETWORKS):

As the second stage of prototype construction, firstly, MDM prototypes are encoded into an embedding space. The conventional few-shot learning approach proposed applying a common convolutional-based architecture to create an embedding space. In contrast, our methodology recommends utilizing transformer-based embedding known as ViT (Dosovitskiy et al., 2020).

After generating an embedded space by a ViT ($f_\phi$), from MDMs ($M_I$) of sample $I$, prototype tensors of each class $C_k$ are created by calculating the mean of embedded tensors, as can be observed in equation 4. Also, $S_k$ is the support set.

$$C_k = \frac{1}{|S_K|} \sum_{(X_I, Y_I) \in S_K} f_\phi(M_I) \tag{4}$$

In the first introduction of prototypical networks, the softmax loss function was recommended over model training; however, we discovered that the categorical cross-entropy loss function leads to more promising outcomes. Therefore, our prototypical network employs a categorical cross-entropy loss function.

The second-stage feature extractor uses episodic meta-learning for training. In every episode of training, the ViT backbone (initialized from ImageNet) gets multiple samples per class (support + query). Class prototypes are computed from support embeddings, the prototypical loss (categorical cross-entropy) on queries is backpropagated to update the backbone with gradients. No classifier head is trained; only the backbone is optimized based on prototype distances.

# 5 EXPERIMENTAL RESULTS

## 5.1 EXPERIMENTAL SETUP

We employed a class-wise rotational split across the six subclasses to ensure that day/night variations are evenly distributed among the training, validation, and test sets. Specifically, 5 folds were implemented, while in each fold, two subclasses are assigned to the test set, two to validation, and two to the training set. That means 2-way classification in few-shot learning. For few-shot episodes, each class provides a support set with either one labelled MDM sample per class in the 1-shot mode or five labelled MDM samples per class in the 5-shot mode. All remaining MDMs in the class are used as query samples in different episodes, and each episode contains 5 query samples. The sampling follows the standard episodic few-shot protocol, which is 1-shot and 5-shot, 2-way classification. In the experiments, we used 1000 episodes for training and 600 episodes for testing. The evaluation criteria are accuracy and Confidence Interval (CI), and due to having such a balanced dataset, we ignored calculating precision, recall and F1-score. All evaluation criteria were selected according to the study of Snell et al (Snell et al., 2017).

## 5.2 OBJECT DETECTION STAGE

In this work, the first object detection section is the first step in inputting effective features into the proposed model. In this stage, Sayeedi et al. (Sayeedi et al., 2024) used the YOLO model to detect objects in biomedical environments linked to mosquitoes. Therefore, in the object detection stage, we implemented various versions of YOLO and newer versions of YOLO. Our results demonstrated that YOLO 11M is able to detect mosquitoes with a 97.8% mAP50, effectively differentiating them from the background. Figure 4-A indicates the images related to the accuracy graph of YOLO 11 and its confusion matrix. Further versions of YOLO implemented in this study are listed in Table 1.

## 5.3 EMBEDDING AND BASELINES

To ensure the superiority of the proposed method, we also compared this model with simple classifiers. The purpose of the implementation was to ensure that after the MDM creation and embedding space, the features obtained cannot be classified with a simple model and require a more complex model. Therefore, a linear baseline and a nearest neighbor classifier were implemented in the final

Table 1: Different YOLO versions performance based on mAP50.

| Yolo Version | mAP50 (%) |
|---|---|
| YOLO 8N | 96.9 |
| YOLO 8M | 96.7 |
| YOLO 11N | 96.5 |
| **YOLO 11M** | **97.8** |

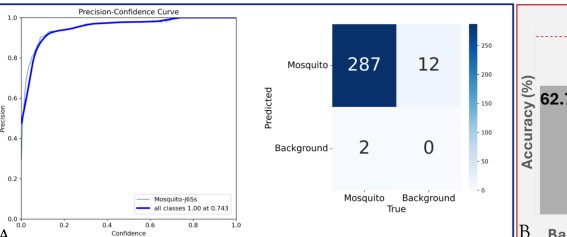
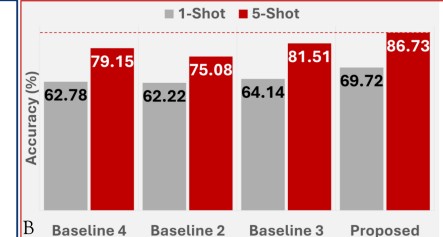

Figure 4: A) Precision-confidence curve and YOLO confusion matrix. B) Performance comparison of three baselines and the proposed two-stage Prototypical Network: Baseline 1 (Linear + cross-entropy), Baseline 2 (Nearest Neighbor + cross-entropy), Baseline 3 (Euclidean + Softmax), and the Proposed method (Euclidean + cross-entropy).

classification stages. Additionally, the baseline method of Snell et al. uses Euclid distance and the Softmax loss function to train the model. In this study, we employed the categorical cross-entropy loss function and demonstrated that utilizing this loss function effectively enhances the learning capabilities of the proposed method. The results are presented in Figure 4-B. In the baseline paper, a CNN was used to build the embedding space. Thus, we first used state-of-the-art pre-trained models, namely EfficientNet V2 (Tan & Le, 2021) and ResNet 101 V2 (He et al., 2016), and the ViT pre-trained model was also utilized. Then it was discovered that this model can extract more effective features from MDMs. Therefore, in the proposed method, the ViT model was used to construct an embedding space. Details can be seen in Table 2.

### 5.4 PROPOSED TWO-STAGE PROTOTYPICAL PERFORMANCE

Table 3 illustrates the model's performance across five folds. In the proposed two-stage prototypical network, training was executed in 1000 episodes, since training became consistent before 1000 episodes. Therefore, we hired checkpoints for validation loss check, to save the best model during the 1000 episodes. Additionally, we selected 600 episodes for testing according to baseline papers. The model was trained in 2-way classification with a query set of 5 and 1, 5 support sets. Due to the biological dataset limitation, we are unable to execute the model in 3-way, 5-way, etc. Overall, the accuracy stayed high, with only moderate differences from one fold to another. Even though a few folds were slightly weaker, the general trend is clear: the model is able to pick up consistent patterns and hold its performance across different splits of the data. The relatively tight confidence intervals also give us confidence that these results are not just a fluke of a single run but reflect a stable underlying behavior of the model.

To assess computational efficiency, inference time of the two heaviest stages in our pipelineYOLO detection and ViT-based embedding extraction were run on T4 GPU 15GB. On this GPU, YOLO ran 32 frames in 0.55 seconds (58 FPS) while backbone ViT took approximately 184 ms per MDM when run in batches. All results are provided in Appendix B. Thus, measurements validate that even though real-time deployment is not achievable with the proposed approach, it can be deemed as an adequate level of computation for offline biological analysis, which is exactly the nature of our case study.

### 5.5 ABLATION STUDIES

First, MDMs were normalized and standardized. In both pre-processing procedures, the model encountered a severe performance decrement of 58.12% in best. Regarding data augmentation, we

Table 2: Performance of different pre-trained models for embedding space.

| Embedding | Accuracy (%) | | Training Time | |
|---|---|---|---|---|
| | 1-Shot | 5-Shot | 1-Shot (h:m:s) | 5-Shot (h:m:s) |
| ResNet | 58.58 | 68.50 | 1:18:29 | 1:19:15 |
| Efficient Net | 58.72 | 71.47 | 1:00:11 | 0:59:20 |
| **ViT** | **69.72** | **86.73** | **0:37:55** | **0:38:42** |

Table 3: Performance of the proposed two-stage prototypical network using MDM prototypes.

| Fold | Accuracy (%) ± CI | |
|---|---|---|
| | 1-Shot | 5-Shot |
| 1 | 69.72 ± 1.53 | 86.73 ± 0.79 |
| 2 | 53.40 ± 1.39 | 66.12 ± 1.14 |
| 3 | 57.00 ± 1.15 | 83.47 ± 0.99 |
| 4 | 70.00 ± 1.33 | 90.73 ± 0.72 |
| 5 | 97.13 ± 0.46 | 98.23 ± 0.33 |
| **Overall** | **69.85** | **85.86** |

increased the number of MDMs before the embedding stage using image processing techniques, including left, right, up, and down shifts, as well as brightness changes. Therefore, we increased the MDM numbers by 5 times, and it was revealed that the data augmentation technique degraded performance. Then, we froze all the initial layers of the ViT model and allowed the model to be trained with only the last two layers. In this case, the model also experienced a severe performance of 50%. As a result, we set all the layers to trainable mode.

In terms of extra dense layers, we added layers with 128 and 256 units to the end of the pre-trained ViT model which was then trained on new data. In this experiment, adding additional layers to the backbone embedding also decreased performance. Consequently, no extra dense layer was added to the backbone. Furthermore, different optimizer functions were applied. According to the outcomes, it was determined that the Adam optimizer provides the best promising results. All details of ablation studies can be found in Tables 4.

Table 5 shows an ablation study over different core design components within the proposed pipeline. The grid-based density map baseline provides marginal performance, showing that coarse spatial aggregation cannot model insect behavior adequately. Replacing the MDM stage with ViT embeddings with simple temporal aggregation, either average or maximum pooling, increases accuracy; max pooling performs best among 1-shot variants. However, all of these baselines are significantly lower than the proposed two-stage prototypical network, which gives 69.72% and 86.73% accuracy in 1-shot and 5-shot setups, respectively. This empirically establishes the fact that explicit modeling of movement dynamics via MDMs, followed by transformer-based embedding and class-level prototyping, is essential towards reliable recognition of mosquito infection behavior under few-shot setups. Full outcomes for the two nearest core designs are provided in the Appendix C.

## 5.6 GENERALIZATION AND ROBUSTNESS

To check how generalizable our proposed method is, we tested it on another dataset. Due to the specific case study, there was no video of mosquitoes flying, so we used MosqutoFusion dataset (Sayeedi et al., 2024), which contains images of mosquitoes. MosquitoFusion is used only for cross-domain generalization and is not part of our few-shot training protocol. In this dataset, we concatenated every 5 images as a short video to get utilizable in our proposed method. This dataset contains three classes including mosquitoes, swarms, and breeding sites. We used mosquito and swarm classes which are linked to our case study. Then, validation and test sets were integrated together to test the proposed method. The results show that although these short videos do not represent the flying behavior of mosquitoes because of the nature of MosquitoFusion dataset, our model is still able to classify videos by generating MDMs and two-stage prototypical networks. Furthermore, since we aim to test the performance of the proposed model in noisy conditions, some normal noise with 0.1 noise factor was added to videos of our proposed dataset. The results illustrate that the model has acceptable robustness to disturbances. Consequently, our findings indicate that the

Table 4: Performance of the proposed method after applying different hyperparameters.

| Ablation | Accuracy (%) ± CI | |
|---|---|---|
| | 1-Shot | 5-Shot |
| Normalization | 58.12 ± 01.13 | 51.43 ± 01.18 |
| Data Augmentation | 60.72 ± 01.53 | 69.68 ± 01.20 |
| Frozen Backbone | 50.0 ± 0.05 | 50.10 ± 0.13 |
| Extra Dense 128 | 52.00 ± 01.18 | 49.45 ± 01.08 |
| Extra Dense 256 | 49.93 ± 01.21 | 48.02 ± 01.09 |
| RMSprop Optimizer | 64.20 ± 01.39 | 82.28 ± 0.94 |
| SGD Optimizer | 67.97 ± 01.34 | 74.47 ± 01.15 |
| **Ours** | **69.72 ± 01.53** | **86.73 ± 0.79** |

Table 5: Performance of the various core design components within the proposed pipeline.

| Ablation | Accuracy (%) ± CI | |
|---|---|---|
| | 1-Shot | 5-Shot |
| Grid Density Map + ProtoNet | 50.00 ± 0.0 | 50.08 ± 0.17 |
| ViT + Average Pooling + ProtoNet | 51.90 ± 1.20 | 60.30 ± 1.07 |
| ViT + Max Pooling + ProtoNet | 77.30 ± 1.05 | 80.60 ± 0.65 |
| **Two-Stage ProtoNet (Ours)** | **69.72 ± 01.53** | **86.73 ± 0.79** |

Table 6: Performance of the proposed method under generalization and robustness tests.

| Generalization Test | Accuracy (%) ± CI | |
|---|---|---|
| | 1-Shot | 5-Shot |
| Cross-Domain by MosquitoFusion Dataset | 57.90 ± 01.32 | 60.87 ± 01.29 |
| Cross-Domain by Noisy Dataset | 67.87 ± 01.41 | 76.17 ± 01.01 |
| Spatial Jigsaw Shuffle Test | 61.27 ± 01.51 | 74.02 ± 0.99 |
| Spatial Blur Test | 58.22 ± 01.34 | 77.83 ± 0.96 |
| Scale Variation Test | 59.93 ± 01.48 | 78.85 ± 0.94 |
| **Original Dataset** | **69.72 ± 01.53** | **86.73 ± 0.79** |

model not only has the ability to generalize to further datasets but also has acceptable robustness in noisy situations. A summary of the results can be seen in Table 6. Since the nature of our case study leads to extremely limited publicly available datasets, showing robustness and generalization bears challenges. Therefore, plus to cross-domain evaluation by MosquitoFusion dataset and the noisy dataset, we implemented three extra tests including spatial jigsaw shuffle, spatial blurring and scale variation on the test set of our own dataset to evaluate robustness and demonstrate generalization. The two-stage prototypical network still provides promising performance, indicating that the model is stable under realistic perturbations of the MDMs and does not overfit to a specific appearance pattern. Outcomes are visible in Table 6.

# 6 CONCLUSION

In this study, we aim to address a straightforward yet often overlooked issue. Conventional AI methods create a wrong embedding space when the behaviour of tiny objects in biomedical videos is the goal of a classification. Our proposed methodology demonstrated that two-stage prototyping for prototypical networks can effectively address this challenge. By turning mosquito movements into MDMs, we created prototypes that are not only compact and biologically sensible but also make a valid embedding space for few-shot classification. The two-stage prototyping we introduced proved to be an effective way of aligning machine learning models with the actual biology of infection-driven behaviour. Across our experiments, this design led to clear improvements in classification accuracy and produced more stable results compared to standard approaches. In summary, our contribution offers both methodological novelty and practical potential, and we hope it will inspire further exploration at the intersection of biomedical research and advanced machine learning techniques.

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

## A APPENDIX

Animal ethics and consent to participate declarations are not applicable for this research as it did not involve humans or animals. The source code of the research is available via the link below: https://github.com/csaiprojects-hub/Two-Stage-Prototypical-Networks-by-Movement-Density-Maps

## B APPENDIX

**Overall efficiency and interface time of the two-stage prototypical network.**

| GPU Type | YOLO per frame(ms) | YOLO FPS | YOLO per 32 frames (s) | Single MDM (ms) | Batched MDM (ms) | Per-MDM in batch (ms) |
|---|---|---|---|---|---|---|
| T4 | 17.14 | 58.34 | 0.55 | 219.40±540.24 | 7360.85 | 184.02 |

## C APPENDIX

**Full results of 5 folds for two nearest core designs: ViT + Average Pooling (AP) + ProtoNet and ViT + Max Pooling (MP) + ProtoNet.**

| Fold | AP Accuracy (%) $\pm$ CI | | MP Accuracy (%) $\pm$ CI | |
|---|---|---|---|---|
| | 1-Shot | 5-Shot | 1-Shot | 5-Shot |
| 1 | 51.9 ± 1.20 | 60.3 ± 1.07 | 77.3 ± 1.05 | 80.6 ± 0.65 |
| 2 | 52.3 ± 0.84 | 52.5 ± 1.02 | 60.4 ± 1.74 | 77.8 ± 0.75 |
| 3 | 51.7 ± 0.55 | 49.4 ± 0.79 | 53.3 ± 0.98 | 60.9 ± 0.81 |
| 4 | 50.4 ± 0.67 | 53.1 ± 0.96 | 58.8 ± 0.82 | 59.7 ± 1.19 |
| 5 | 55.4 ± 1.09 | 69.1 ± 0.94 | 67.9 ± 1.75 | 84.7 ± 0.67 |
| **Overall** | **52.34** | **56.88** | **63.54** | **72.74** |

