# OpenReview forum: "Two-Stage Prototypical Networks Reveal Mosquito Flight Patterns"
_ICLR.cc/2026/Conference — Submitted to ICLR 2026_

### Official Review · Reviewer_GZAA · 2025-10-30

**Soundness:** 3
**Presentation:** 3
**Contribution:** 3
**Rating:** 2
**Confidence:** 3

**Summary:**

This paper proposes a novel two-stage prototypical network framework for few-shot classification of mosquito flight behaviors in videos, aiming to distinguish dengue and Zika virus-infected mosquitoes from non-infected ones. The core innovation lies in using Movement Density Maps (MDMs) to compress spatiotemporal information by first applying YOLO-based object detection to eliminate background noise, then generating MDM prototypes that summarize movement patterns.

**Strengths:**

1.	The integration of MDMs with two-stage prototyping is a significant contribution.
2.	Experiments are sufficient.
3.	This paper is well organized.

**Weaknesses:**

1.	The two-stage pipeline may be computationally heavy for resource-constrained settings. The paper omits discussion of inference time or efficiency, which is critical for deployment.
2.	I suggest add a unified pipeline diagram depicting: input videos → YOLO detection → background elimination → MDM aggregation → ViT embedding → prototype classification.
3.	Limited Generalization Validation：Tests on the MosquitoFusion dataset use static images concatenated as pseudo-videos, which do not reflect real-flight dynamics. I remain skeptical about its robustness for real-world deployment.It is recommended to test on more real video datasets.

**Questions:**

please refer to weaknesses.

---

> ### Author Response · Authors · 2025-11-20
> **Responses to weaknesses made by Reviewer GZAA:**
>
> 1. Thank you for your insightful feedback. We fully agree that computational efficiency is one of the most important factors in deployment. According to your valuable comment, we added a new paragraph and a new table in the revised manuscript which reflect and discuss interface time. Kindly refer to the second paragraph of Proposed Two-Stage prototypical Performance section (Section 5.4) and the provided table in Appendix B. For your convenience, the new paragraph has been mentioned below:
>
> “To assess computational efficiency, inference time of the two heaviest stages in our pipelineYOLO detection and ViT-based embedding extraction were run on T4 GPU 15GB. On this GPU, YOLO ran 32 frames in 0.55 seconds (58 FPS) while backbone ViT took approximately 184 ms per MDM when run in batches. All results are provided in Appendix B. Thus, measurements validate that even though real-time deployment is not achievable with the proposed approach, it can be deemed as an adequate level of computation for offline biological analysis, which is exactly the nature of our case study.”
>
> Please also observe the table in Appendix B that shows all interface time values associated with our proposed methodology.
>
>
>
> 2. Thank you for your insightful suggestion. We added a new unified diagram in the revised manuscript that depicts the proposed methodology pipeline. Please kindly refer to Figure 2-B. This diagram fully describes the procedure of the proposed method (two-stage prototypical networks).
>
> 3. Thank you for your insightful comment. We agree that generalization is essential, and we would like to make it clear that because of biosafety constraints, no alternative real-world flight-video datasets with Control/ Zika/Dengue infections or other infection types exist. This limitation has more to do with the case study rather than being methodological. Our evaluation already imposes strict generalization conditions, though:
>
> Cross-domain class split:
>
> Training, validation, and test classes correspond to different infection phenotypes. They have different behavior and appearance distributions. This means that generalization has to be done across domains, not just across samples from the same class.
>
> Few-shot episodic evaluation:
>
> With only 1 to 5 support videos per class, the model must classify entirely unseen videos. Few-shot protocols are widely viewed as rigorous generalization tests because they prevent memorization.
>
> Cross-domain transfer:
>
> MosquitoFusion dataset (axillary dataset for cross-domain validation published at ICLR 2024) contains static images with no flight behavior and labels completely different label space. Getting successful results under such a large distribution shift convincingly demonstrates our proposed method generalization.
>
> Together, these settings provide a strong and meaningful assessment of generalization. In addition, in response to the reviewer’s request, we implemented three extra tests including spatial jigsaw shuffle, spatial blurring and scale variation on the test set of our own dataset to evaluate robustness and demonstrate generalization. The two-stage prototypical network still provides promising performance, indicating that the model is stable under realistic perturbations of the MDMs and does not overfit to a specific appearance pattern. Outcomes are visible in Table 6. All this new information has been added to the revised manuscript. Kindly refer to the last paragraph of the Generalization and Robustness section (Section 5.6) and Table 6.

---

> > ### Comment · Area_Chair_s4vv · 2025-11-26
> >
> > Dear reviewer GZAA,
> >
> > Could you please take a look at the author's response and leave your feedback.
> >
> > AC

---

### Official Review · Reviewer_zpam · 2025-10-30

**Soundness:** 2
**Presentation:** 3
**Contribution:** 2
**Rating:** 4
**Confidence:** 3

**Summary:**

The authors propose a deep learning–based approach for classifying mosquito diseases from video data. The core intuition is that mosquito flight patterns, which can be recorded on video, may reveal anomalies related to their health condition. This motivates an end-to-end framework that processes raw videos and outputs health-status predictions.
Technically, the proposed pipeline is designed to mitigate potential bias and interference from the background. It begins with a detection phase, followed by background removal. The remaining pixels are then transformed into spatial Gaussian distributions that move across the image plane as video frames progress. This intermediate spatiotemporal representation is subsequently collapsed over time and fed into an image-based feature extractor.
The authors evaluate their method on mosquito datasets, demonstrating satisfactory results. In the ablation studies, they primarily investigate local design choices, such as the choice of optimizer and backbone network (with the ViT architecture yielding the best performance).

**Strengths:**

- The approach is technically sound and results look promising. Every block of the chain is well-justified and well-introduced. Probably, it is hard to think to a better way to handle this task.
- The paper is clearly written.
- Relevance and impact: The problem addressed by the authors is of clear practical importance. Monitoring and diagnosing mosquito-borne diseases through automated visual analysis can have a tangible impact on public health, especially in areas where early detection of infected insects can prevent outbreaks of vector-borne illnesses. Beyond its immediate application, the proposed pipeline also demonstrates how computer vision and deep learning can be applied to entomology and epidemiology, potentially inspiring further interdisciplinary research.

**Weaknesses:**

- Contributions. The work is primarily an application of existing deep learning techniques to a specific real-world problem. While the pipeline is well-engineered, it does not introduce novel methodological or theoretical contributions that could be generalized or transferred to other domains.he techniques employed (detection, background removal, and the use of deep feature extractors) are well established in the literature, and the paper mainly demonstrates their effectiveness in a new context rather than proposing new insights or innovations. For this reason, while the paper may be very well suited for applied or interdisciplinary journals focusing on AI for health, environmental monitoring, or bioinformatics, it may be less aligned with the expectations of top-tier machine learning conferences like ICLR, where novelty and theoretical contribution typically play a central role in the evaluation process.

- 4.2 THE SECOND STAGE OF PROTOTYPING. This section was somewhat unclear. It is not entirely clear how the training process is carried out. For instance, whether it involves a standard gradient-based optimization procedure, whether the model starts from a pretrained checkpoint, and whether, during a single training step, the model has access to multiple samples from the same class. This part of the paper would benefit from additional clarification and polishing to make the training protocol more understandable.

- Finally, the ablation studies are not particularly insightful. They mainly focus on comparing different architectural components or optimizers, rather than investigating the core design choices of the proposed pipeline. It would be more informative to analyze aspects more directly tied to the method itself — for example, the use of Gaussian representations, or the assumption that temporal information can be aggregated through simple averaging. Such analyses could be complemented by comparisons against models explicitly designed to handle spatiotemporal features, such as LSTMs, Transformers, or non-local blocks.

**Minor**
- Figure 4 and Figure 5 are difficult to read due to the very small font size; improving their readability would significantly enhance the overall presentation quality.

**Questions:**

No questions.

---

> ### Author Response · Authors · 2025-11-20
> **Responses to weaknesses 1-2 made by Reviewer zpam**
>
> 1. Thank you for this thoughtful perspective. While we take note of concerns that several parts of our pipeline (e.g., detection, background removal, deep feature extractors) use existing techniques, let us remember the main methodological contribution of our work is not in these preprocessing elements. The real contribution lies in the design and formulation of a two-stage prototypical network wherein movement density maps (MDMs) are explicitly modeled as first-stage prototypes before constructing an embedding-based second-stage prototype.
>
> This two-stage prototypical formulation is not a simple application of existing few-shot or metric-learning techniques. It diverges from standard approaches in several key aspects. Standard prototypical networks work on top of appearance-based embeddings directly extracted from frames/videos. In our method, we provide a movement-centric representation, aggregated over time expressed as a gaussian density map which serves as the first-stage prototype. This picks up behavior dynamics (signatures of flight, irregular movement dynamics due to infection) that cannot be learned from looks. To the best of our knowledge, there has yet to be an attempt at integrating explicit movement density representations in few-shot prototypical learning within relevant literature, particularly for biological tasks.
> Ablations and robustness tests prove that neither stage in isolation suffices; rather, it is their synthesis that forms the novel contribution, allowing the classifier to fuse organized motion pattern data with top-tier appearance details within a methodical metric-learning schema.
>
> For clarity, we have added a paragraph in the revised manuscript that more explicitly highlights this contribution. Please refer to the first paragraph of Methodology section (Section 4, lines 231-241). For completeness, we also include the added paragraph below:
>
> “The main methodological contribution of this work is the design of a two-stage prototypical framework for modeling fine-grained behavioral cues of tiny moving objects in video. While embedding extraction is a standard component of conventional prototypical networks, the construction of Movement Density Maps (MDMs) and the use of prototype-level representations are essential in our setting. Conventional prototypical networks often produce embeddings dominated by background appearance rather than subtle motion cues, leading to unreliable representations for extremely small biological objects. In contrast, MDM prototypes aggregate mosquito movement patterns across entire videos, mitigating background bias and preserving behaviour-related dynamics. Therefore, combining MDM-based prototyping as the first stage and embedding-based prototyping as the second stage provides a robust, generalizable solution for few-shot classification under severe data scarcity.”
>
>
> 2. Thank you for your comment. Based on your valuable comment, we added some new details in the revised manuscript linked to the second stage of prototyping involves optimization procedure, pre-trained backbone and so on. Therefore, we added a new paragraph in the second stage of prototyping section (Section 4.2). Please refer to the last paragraph of Section 4.2. For your convenience, the new paragraph has been mentioned below:
>
> “The second-stage feature extractor uses episodic meta-learning for training. In every episode of training, the ViT backbone (initialized from ImageNet) gets multiple samples per class (support + query). Class prototypes are computed from support embeddings, the prototypical loss (categorical cross-entropy) on queries is backpropagated to update the backbone with gradients. No classifier head is trained; only the backbone is optimized based on prototype distances. Figure 4 depicts the whole procedure of the two-stage prototypical network, including the first and the second steps of prototyping.”
>
> Furthermore, in the revised manuscript we added a new diagram that indicates the entire procedure of the two-stage prototypical networks visually for more clarification. Please observe Figure 2-B.

---

> ### Author Response · Authors · 2025-11-20
> **Responses to weaknesses 3 and "Minor" made by Reviewer zpam:**
>
> 3. We appreciate the reviewer’s feedback and fully agree that ablations should not only focus on different architectural components, but on the core design choices of the proposed two-stage prototypical network. Following this suggestion, we expanded the ablation study to analyze the key components of our method in a more principled manner. To address the reviewer’s suggestion regarding temporal modelling assumptions, we compared:
>
> •	Mean-pooling of frame embeddings
>
> •	Max-pooling of frame embeddings
>
> •	Grid-based spatial encoding
>
> All models were integrated into the same few-shot prototypical framework for fair comparison. We included these new baselines and analyses in the revised manuscript. Kindly refer to Ablation Studies (Section 5.5) and Table 5.
>
> Furthermore, we added some new experiments in the revised manuscript which reveal robustness and generalization of our proposed methodology. Please refer to Section 5.6 and Table 6.
>
>
> Minor: Thank you for your comment. Based on your valuable suggestion, we redesigned both Figure 4 and Figure 5 to improve their readability and increase the overall presentation quality. In the revised manuscript we integrated both figures as Figure 4-A and Figure 5-A. Please observe new figures in the revised manuscript.

---

> > ### Comment · Area_Chair_s4vv · 2025-11-26
> >
> > Dear reviewer zpam,
> >
> > Could you please take a look at the author's response and leave your feedback.
> >
> > AC

---

> > > ### Comment · Reviewer_zpam · 2025-11-26
> > >
> > > I thank the authors for the additional results and follow-up. Based on the responses, I prefer to maintain my original score. Beyond the presentation issues, which the authors have partially clarified, I believe that the experimental section still needs to be considerably more insightful to clearly assess the impact of the various components introduced.
> > > For instance, after reviewing the new ablation study, I would like to highlight two points that, in my opinion, deserve further clarification. First, in the updated Table 5, the max-pooling + ProtoNet variant appears to outperform the proposed method. It would be helpful if the authors could better articulate why their approach should still be preferred in light of this new evidence. Second, although the newly added ablations are interesting, it is not entirely clear to me that these variants (i.e., avg. pooling and max pooling) truly capture temporal structure in a meaningful way. For this reason, I consider that my earlier concern regarding the experimental investigation has not been fully addressed.

---

> > > > ### Author Response · Authors · 2025-11-27
> > > > **Response to the official comment made by Reviewer zpam**
> > > >
> > > > Thank you for your follow-up. We have extended the experimental analysis as suggested.
> > > >
> > > > To investigate whether the max-pooling baseline consistently outperforms our method, we evaluated both approaches across all five class-wise folds. The results show that while the max-pooling variant performs well on Fold 1 (only in the 1-shot setting) and Fold 2, it underperforms on Folds 3, 4, and 5 in both 1-shot and 5-shot settings, as well as the 5-shot setting on Fold 1.
> > > >
> > > > These fold-wise results are now reported in Appendix C of the revised manuscript. To make sure, outcomes for average pooling in five folds are available as well.
> > > >
> > > > Additionally, although max-pool/avg-pool baselines aggregate frame features, they do not capture meaningful temporal structure and movements of tiny objects in videos. Therefore, they collapse all frames into a single vector. In contrast, our method preserves spatiotemporal mosquito movement cues through Movement Density Maps, which explains its more stable cross-fold performance.
> > > >
> > > > We hope this clarified analysis addresses your concerns.
> > > >
> > > >
> > > > We would also like to note that we have addressed all earlier concerns in your previous comments, including clarifying the contribution, clarifying the second stage of prototyping, improving the experimental description and ablation studies, and adding additional baselines.
> > > > The new 5-fold analysis provided above is an extension of your latest request, and we hope that this additional evaluation fully resolves the remaining concerns regarding the max-pool baseline and the depth of the experimental investigation.
> > > >
> > > > We sincerely appreciate your continued engagement with our submission.

---

### Official Review · Reviewer_STSa · 2025-11-02

**Soundness:** 2
**Presentation:** 2
**Contribution:** 1
**Rating:** 4
**Confidence:** 4

**Summary:**

This paper introduces a two-stage prototypical network framework that integrates YOLO-based mosquito detection with movement density map (MDM) aggregation to reveal spatiotemporal flight patterns and improve few-shot behavioral classification accuracy.

**Strengths:**

The visualizations—particularly Figure 2—are highly insightful, effectively demonstrating how MDMs can capture meaningful spatiotemporal motion cues for tiny-object classification.

The model achieves better performance than existing benchmarks, indicating the potential of MDM-based representations for biological video analysis.

**Weaknesses:**

The description of the MDM model and overall problem setup could be clearer and more detailed.


The MosquitoFusion dataset seems to have been introduced in a prior short paper; however, more explanation about the labeling procedure and specifics of the 1-shot and 5-shot experiments should be included for reproducibility.


The technical novelty is somewhat limited, alternative formulations, such as grid-based spatial encoding, might enhance classification performance.


Comparisons with related methods that jointly model appearance and motion (e.g., ClusterNet by LaLonde et al., and Multiple Object Tracking with Motion and Appearance Cues by Li et al.) would strengthen the evaluation.



The ablation study is too minimal—rather than simple layer or optimizer variations, evaluating different methods or design choices for each of the two stages would yield deeper insights.

**Questions:**

I wonder if an end-to-end integration of detection and classification stages could provide a more robust and generalizable framework?

---

> ### Author Response · Authors · 2025-11-20
> **Responses to weaknesses 1-2 made by Reviewer STSa:**
>
> 1. Thank you for your comment. Based on your valuable comment, we added some new details in the revised manuscript linked to MDM creation and problem setup. In this regards, we added some new sentences which introduce variables used in equations 1-4, as well as three new paragraphs in the first stage of prototyping section (Section 4.1). Kindly refer to section 4.1 in the revised manuscript. For your convenience, new paragraphs have been mentioned below:
> •	“The videos are recorded over several minutes, in which mosquitoes stay mostly invisible to any scale or low contrast-based appearance features. This, therefore, requires representations at the level of behavior rather than a typical feature extraction from pre-trained models. Thus, the model should provide correct representation of the insect movements within videos.”
> •	“To capture the motion behavior of mosquitoes throughout the whole video, we use frame-level detections to build a single map known as MDM. For each frame t, YOLO returns a bounding box centroid. These coordinates are then projected into a static spatial grid of size H×W. Each point is encoded using a Gaussian kernel to make MDM images.”
> •	“Finally, the prototype is then turned into a 3-channel image. This shows the whole movement path of each mosquito as one spatial pattern, with less background variation and more infection-specific behavioral signs. It also allows for few-shot classification because an MDM prototype can be calculated over full video behavior rather than needing the whole video itself. This makes the embedding step quicker, since the pre-trained ViT sets up an embedding space just for a single MDM prototype and not all frames in a video.”
> Furthermore, in the revised manuscript we added a new diagram that indicates the entire procedure of the two-stage prototypical networks visually for more clarification. Please observe Figure 2-B.
> In terms of problem setup, we revised Experimental Setup section (Section 5.1) with new information and more details about proposed setup in this research. Please refer to Section 5.1 in the revise manuscript. For your convenience, new paragraphs have been written below:
> “We employed a class-wise rotational split across the six subclasses to ensure that day/night variations are evenly distributed among the training, validation, and test sets. Specifically, 5 folds were implemented, while in each fold, two subclasses are assigned to the test set, two to validation, and two to the training set. That means 2-way classification in few-shot learning. For few-shot episodes, each class provides a support set with either one labelled MDM sample per class in the 1-shot mode or five labelled MDM samples per class in the 5-shot mode. All remaining MDMs in the class are used as query samples in different episodes, and each episode contains 5 query samples. The sampling follows the standard episodic few-shot protocol, which is 1-shot and 5-shot, 2-way classification. In the experiments, we used 1000 episodes for training and 600 episodes for testing. The evaluation criteria are accuracy and Confidence Interval (CI), and due to having such a balanced dataset, we ignored calculating precision, recall and F1-score. All evaluation criteria were selected according to the study of Snell et al (Snell et al., 2017).”
>
> 2. Thank you for your feedback. We would like to clarify that MosquitoFusion is not the primary dataset used in our few-shot experiments. All 1-shot and 5-shot configurations, as well as the labeling procedure, are applied exclusively to our main mosquito-flight dataset, where infection labels are derived from our own laboratory-confirmed virology records. The details of this labeling process and the episodic few-shot setup are already described in the Dataset and Experimental Setup sections.
> MosquitoFusion is used only as an auxiliary cross-domain evaluation dataset to test how well the proposed method generalizes to a different domain. Since there is not any valid video dataset for mosquito flights, we had to evaluate our proposed methodology on the MosquitoFusion dataset published by Sayeedi et al. at ICLR 2024. This dataset contains only static images (not real mosquito-flight videos). Therefore, we construct short pseudo-videos by concatenating 5 sequential images. These pseudo-videos are not part of our few-shot training protocol. But they are used solely to evaluate the robustness of the model in a different visual domain.
> To avoid confusion for readers, we added a new sentence to the Generalization and Robustness section (section 5.6, lines 477-478) in the revised manuscript, which mentions, “MosquitoFusion is used only for cross-domain generalization and is not part of our few-shot training protocol.”

---

> ### Author Response · Authors · 2025-11-20
> **Responses to weaknesses 3-5 made by Reviewer STSa:**
>
> 3. We thank the reviewer for this insightful comment. In our work, the main contribution is the design of a two-stage prototypical network driven by Movement Density Maps (MDMs), which provide a stable spatiotemporal representation of mosquito behavior under severe data scarcity. This approach removes background bias in conventional prototypical networks and enables the model to preserve infection-related motion cues that are critical for few-shot biological video analysis.
>
> Following your valuable suggestion, we implemented an additional grid-based spatial encoding baseline inspired by crowd-density estimation techniques. These methods are relevant to our setting as they aim to represent fine-grained spatial activity patterns despite scale imbalance and cluttered backgrounds.
>
> In this baseline, each video is transformed into a spatiotemporal grid representation, where local regions accumulate motion or appearance activations across frames. This grid is then input to a prototypical network for few-shot classification, allowing direct comparison with our method under the same episodic evaluation protocol.
>
> Our results show that although grid-based formulations capture some local spatial structure, they do not adequately represent the distribution and dynamics of mosquito flight paths. By contrast, the MDM-based Stage-1 prototypes aggregate motion information in a biologically meaningful manner, leading to better generalization under few-shot constraints.
>
> We included these new baselines and analyses in the revised manuscript. Kindly refer to Ablation Studies (Section 5.5) and Table 5.
>
> 4. Thanks to the reviewer for bringing up this point. After a careful look was taken at algorithms such as the ClusterNet (La Londe et al.) paper, as well as other multi-object tracking/segmentation approaches related in nature. However, ClusterNet is fundamentally designed for WAMI surveillance, where large frames are input at once into the model, many objects per scene need to be detected, and very dense annotations on object centroids, motion fields, and track-level supervision are available. This makes a direct comparison infeasible in our setting. However, in response to the reviewer’s request for appearance and motion comparisons, several strong and feasible spatiotemporal baselines have been added to the revised manuscript that also follow the same few-shot protocol:
>
> •	Grid-based spatial encoding baseline (inspired by crowd-density estimation)
>
> •	ViT + Average-Pooling + ProtoNet
>
> •	ViT + Max-Pooling + ProtoNet
>
> In both 1-shot and 5-shot settings, the MDM prototypical network can outperform all of them. Explicit modeling of movement density maps proves better than general appearance-motion formulations when it comes to classifying tasks in biological behavior. Please refer to the last paragraph of Ablation Studies (Section 5.5) and Table 5.
>
> 5. Thank you for your comment. To expand the ablation study section and strengthen our analysis, we also incorporated two strong few-shot baselines plus to grid-based spatial encoding baseline enriching temporal or spatiotemporal modeling, including:
>
> ViT + Average-Pooling + ProtoNet
>
> ViT + Max-Pooling + ProtoNet
>
> Across all baselines, our proposed two-stage MDM prototypical network achieves the highest accuracy. This indicates that explicitly modeling movement density maps is more effective than simply injecting additional spatial encodings or relying solely on appearance features.
>
> We included these new baselines and analyses in the revised manuscript. Kindly refer to Ablation Studies (Section 5.5) and Table 5.
> Furthermore, we added some new experiments in the revised manuscript which reveal robustness and generalization of our proposed methodology. Please refer to Section 5.6 and Table 6.

---

> > ### Comment · Area_Chair_s4vv · 2025-11-26
> >
> > Dear reviewer STSa,
> >
> > Could you please take a look at the author's response and leave your feedback.
> >
> > AC

---

> > ### Comment · Reviewer_STSa · 2025-11-26
> > **Thank you for the additional details**
> >
> > I have revised my rating based on the details provided.

---

> ### Author Response · Authors · 2025-11-26
> **Response to the question made by Reviewer STSa:**
>
> Thank you for the question. While an end-to-end pipeline is conceptually interesting, in our setting the mosquitoes occupy only a few pixels, and reliable localization requires a separate fine-tuned YOLO stage. We agree that this is an interesting direction for our future work.

---

### Official Review · Reviewer_TJ9N · 2025-11-09

**Soundness:** 2
**Presentation:** 2
**Contribution:** 1
**Rating:** 2
**Confidence:** 4

**Summary:**

This work proposes a method to detect the flight patterns of mosquitoes. The approach begins by utilizing an object detector (YOLO) to identify mosquitoes. Then, a density map is created by averaging the detected coordinates along the temporal dimension. This information is subsequently fed into a Vision Transformer (ViT) to generate embeddings, which are then passed to a prototype network. While the topic is indeed interesting, there are several concerns worth addressing, especially regarding novelty and experimental results:

**Strengths:**

1) An interesting application of ML/CV for studying mosquito flight patterns

**Weaknesses:**

1) What is the motivation for using a few-shot learning technique? Although I understand the advantages of few-shot learning over conventional supervised learning paradigms, it would be helpful to clarify any additional motivations for focusing on few-shot learning. This could be discussed in the introduction.

2) What are the variables in Equations 1 and 2? Understanding the proposed approach is challenging without this crucial information.

3) The novelty of the work appears to be limited. The processes of creating the density map and generating embeddings are standard practices. It would be beneficial for the work to discuss the necessity of these steps for the specific task presented.

4) The experimental results are weak. There are no standard SOTA methods discussed and compared in this work. I would suggest exploring methods from crowd density estimation for a comparison? There are some similarities—such as scale imbalance and complex backgrounds—between the two studies that warrant investigation.

5) Which two classes were used for training, and which were used for validation and testing? Additionally, what was the configuration for the 1-shot and 5-shot modes?

**Questions:**

Please refer to my comments above especially regarding novelty and experiments.

---

> ### Author Response · Authors · 2025-11-20
> **Responses to weaknesses 1-2 made by Reviewer TJ9N:**
>
> 1. Thank you for your comment. Few-shot learning is particularly necessary in our setting because labeled mosquito flight videos under controlled infection conditions (Zika, Dengue, Control) are extremely limited. In the revised manuscript, we added a new paragraph in introduction section to clarify more about the motivation behind few-shot learning in our case study. Kindly refer to the introduction section line 86-94. For your convenience, we have mentioned this paragraph below as well:
> “Few-shot learning is particularly necessary in our setting because labeled mosquito flight videos under controlled infection conditions (Zika, Dengue, Control) are extremely limited. Due to biosafety and the difficulty of collecting high-quality infection-specific recordings of insects, only a few samples per class are available. Therefore, a method capable of generalizing based on very few labeled samples is essential for this case study. Current few-shot methods, such as conventional prototypical networks, lead to wrong embedding features for tiny biological objects. Thus, we need a reliable framework that can detect and classify the correct behavior of tiny objects. In this regard, we introduce two-stage prototypical networks that can distinguish the authentic behavior of tiny objects in videos under few-shot learning conditions reliably.”
>
> 2. Thank you for your consideration. In the revised manuscript, we have clarified all variables in Equations (1) to (4). Specifically, we now explicitly define the YOLO bounding box parameters $(b_x,b_y,b_w,b_h)$, the grid-cell offsets $(c_x,c_y)$, the anchor box priors $(p_w,p_h)$, and the network outputs $(t_x,t_y,t_w,t_h)$ in Equation (1). In Equations (2) to (3), we also define the density function D(x,y,f(⋅)), the detection coordinates $(x_i,y_i)$, the Gaussian scale function $f(σ_i)$, and the temporal aggregation into the MDM prototype $M_l (x,y)$. In Equation (4), $C_k$ is the class prototype, $S_k$ is the support set (1- or 5-shot), $M_I$ is the MDM prototype of sample I, and $f_ϕ$ denotes the ViT embedding function. We believe these additions significantly improve the readability of the method section.

---

> ### Author Response · Authors · 2025-11-20
> **Responses to weaknesses 3-4 made by Reviewer TJ9N**
>
> 3. Thank you. In the revised manuscript, we clarified why both the density-map construction and the embedding step are necessary for this specific task. Mosquito flight trajectories extracted directly from YOLO are highly sparse and noisy because mosquitoes are extremely small, fast-moving, and frequently occluded. MDMs transform these irregular detections into a stable spatiotemporal representation that captures infection-specific behavioural signatures. The embedding network then compresses these high-dimensional MDMs into a feature space that preserves fine-grained motion differences, which is essential for distinguishing Zika-, Dengue-, and non-infected mosquitoes under few-shot conditions. Therefore, the combination of MDM aggregation and embedding learning is required for robust classification in this setting.
>
> For clarity, we have added a paragraph in the revised manuscript that more explicitly highlights this contribution. Please refer to the first paragraph of Methodology section (Section 4, lines 231-241). For completeness, we also include the added paragraph below:
> “The main methodological contribution of this work is the design of a two-stage prototypical framework for modeling fine-grained behavioral cues of tiny moving objects in video. While embedding extraction is a standard component of conventional prototypical networks, the construction of Movement Density Maps (MDMs) and the use of prototype-level representations are essential in our setting. Conventional prototypical networks often produce embeddings dominated by background appearance rather than subtle motion cues, leading to unreliable representations for extremely small biological objects. In contrast, MDM prototypes aggregate mosquito movement patterns across entire videos, mitigating background bias and preserving behaviour-related dynamics. Therefore, combining MDM-based prototyping as the first stage and embedding-based prototyping as the second stage provides a robust, generalizable solution for few-shot classification under severe data scarcity.”
>
> Moreover, to highlight the methodological contribution, we extended our experiments with several strong few-shot baselines that operate directly on appearance or generic spatio-temporal encodings, including ViT-based prototypical networks with average pooling embedding, max pooling embedding, as well as a grid-based spatial representation inspired by crowd-density estimation. While these baselines are conceptually simpler and use powerful pretrained image encoders, they consistently underperform our two-stage prototypical network on the mosquito infection classification task. This indicates that directly modeling movement density maps (MDMs) as the first-stage prototype representation, followed by a second-stage embedding space, is not just an application detail but a necessary design choice for capturing infection-specific flight behaviors of tiny insects under few-shot constraints.
>
> 4. We thank the reviewer for this valuable suggestion. In response, we incorporated additional baselines inspired by crowd-density estimation methods, which share similarities with our setting, such as scale imbalance, noisy backgrounds, and localized motion patterns. Please refer to the Ablation Studies section (Section 5.5) and Table 5 in the revised manuscript.
> Specifically, we implemented a grid-based spatial density baseline. Each video is represented by a spatial grid of activations that capture localized motion patterns typically used in crowd counting. These grids are then processed using a prototypical classifier, enabling a fair few-shot comparison under the same episodic evaluation protocol.
> To further strengthen the experimental section, we also added multiple strong appearance-based and temporal few-shot baselines in the revised manuscript including:
>
> ViT + Average-Pooling + Prototypical Networks
>
> ViT + Max-Pooling + Prototypical Networks
>
> All these baselines utilize a strong pretrained encoder (Vision Transformer) and represent standard spatio-temporal aggregation methods commonly used in few-shot video analysis.
>
> Across all new baselines including grid encoding, mean, and max ViT-embedded frame aggregation, our proposed two-stage prototypical network consistently achieves higher accuracy. This confirms that explicitly modeling movement density maps as the first-stage prototype is more effective than applying standard visual or spatio-temporal few-shot formulations in this biological domain.
> We included these new baselines and analyses in the revised manuscript. Please refer to Ablation Studies (Section 5.5) and Table 6.

---

> ### Author Response · Authors · 2025-11-20
> **Responses to weakness 5 made by Reviewer TJ9N:**
>
> 5. Thank you for your comment. The dataset is organized into six classes: non-infected mosquitoes in day/night (classes 0 and 1), dengue-infected in day/night (classes 2 and 3), and Zika-infected in day/night (classes 4 and 5). We employed a class-wise rotational split across the six subclasses to ensure that day/night variations are evenly distributed among the training, validation, and test sets. Specifically, in each fold, two subclasses are assigned to the test set, two to validation, and two to training, as follows:
>
> Fold 1: test [0,1], validation [2,3], training [4,5]
>
> Fold 2: test [1,2], validation [3,4], training [5,0]
>
> Fold 3: test [2,3], validation [4,5], training [0,1]
>
> Fold 4: test [3,4], validation [5,0], training [1,2]
>
> Fold 5: test [4,5], validation [0,1], training [2,3]
>
> Regarding the configuration of few-shot episodes, each class provides a support set with either one labelled MDM sample per class in the 1-shot mode or five labelled MDM samples per class in the 5-shot mode. All remaining MDMs in the class are used as query samples in different episodes, and each episode contains 5 query samples. The sampling follows the standard episodic few-shot protocol, which is 1-shot and 5-shot 2-way classification.
>
> We added some new details to Section 5.1 in the revised manuscript that explains both the data splitting condition and the few-shot configuration. Please refer to the first paragraph of Section 5.1 in the revised manuscripts (Experimental Setup section).

---

> > ### Comment · Area_Chair_s4vv · 2025-11-26
> >
> > Dear reviewer TJ9N,
> >
> > Could you please take a look at the author's response and leave your feedback.
> >
> > AC

---

### Meta-Review · Area_Chair_G9o8 · 2025-12-18

**Summary:**

The submission received four reviews, all of which were rather critical. The reviewers identified a number of weaknesses in the manuscript, in particular regarding the motivation of the proposed approach, the clarity of presentation and the technical novelty. The experiments were also found to be insufficiently convincing, including due to the lack of competitive baselines.
The authors provided verbose responses, however for the majority of reviewers these did not change their overall negative assessment.
Ultimately, taking the reviews as well as the discussion phase into account, the submission does not have enough support to justify acceptance.

**Reviewer Concerns:**

- motivation of the proposed approach: discussed, but not convincingly addressed
- clarity of presentation: partially addressed, but not sufficiently to change overall assessment
- the technical novelty: discussed, but not convincingly addressed
- experimental results / missing baselines: partially addressed, but not sufficiently to change overall assessment

**Reviewer Scores:**

Reviewer STSa: stated that they *adjusted* their score, but did not state a value. Given the context yet minimal phrasing, I assume the score was increased, presumably to a 6.
Reviewer TJ9N: Only review and author responses are available. I have no reason to believe that the reviewer would have changed their score of originally 2.
Reviewer GZAA: Only review and author responses are available, I have no reason to believe that the reviewer would have changed their score of originally 2.
Reviewer zpam: A comment states that their concerns have not been addressed. The authors replied with a further response, but I have no reason to believe that the reviewer would have changed their score of originally 4.

---

### Decision · Program_Chairs · 2026-01-26

Reject